# External Workload Indicators of Muscle and Kidney Mechanical Injury in Endurance Trail Running

**DOI:** 10.3390/ijerph16203909

**Published:** 2019-10-15

**Authors:** Daniel Rojas-Valverde, Braulio Sánchez-Ureña, José Pino-Ortega, Carlos Gómez-Carmona, Randall Gutiérrez-Vargas, Rafael Timón, Guillermo Olcina

**Affiliations:** 1Centro de Investigación y Diagnóstico en Salud y Deporte, Escuela Ciencias del Movimiento Humano y Calidad de Vida, Universidad Nacional, Heredia 86-3000, Costa Rica; randall.gutierrez.vargas@una.cr; 2Grupo Avances en Entrenamiento Deportivo y Acondicionamiento Físico (GAEDAF), Facultad de Ciencias del Deporte, Universidad de Extremadura, 10005 Cáceres, Spain; rtimon@unex.es; 3Programa de Ciencias del Ejercicio y la Salud, Escuela Ciencias del Movimiento Humano y Calidad de Vida, Universidad Nacional, Heredia 86-3000, Costa Rica; 4Departamento de Actividad Física y Deporte, Facultad de Ciencias del Deporte, Universidad de Murcia. San Javier, 30720 Murcia, Spain; 5Grupo en Optimización del Entrenamiento y Rendimiento Deportivo (GOERD), Facultad de Ciencias del Deporte, Universidad de Extremadura, 10005 Cáceres, Spain; cdgomezcarmona@unex.es

**Keywords:** principal component analysis, mountain sport, acute kidney injury, acute renal failure, exertional rhabdomyolysis

## Abstract

Muscle and kidney injury in endurance athletes is worrying for health, and its relationship with physical external workload (eWL) needs to be explored. This study aimed to analyze which eWL indexes have more influence on muscle and kidney injury biomarkers. 20 well-trained trail runners (age = 38.95 ± 9.99 years) ran ~35.27 km (thermal-index = 23.2 ± 1.8 °C, cumulative-ascend = 1815 m) wearing inertial measurement units (IMU) in six different spots (malleolus peroneus [MP_left_/MP_right_], vastus lateralis [VL_left_/VL_right_], lumbar [L_1_–L_3_], thoracic [T_2_–T_4_]) for eWL measuring using a special suit. Muscle and kidney injury serum biomarkers (creatin-kinase [sCK], creatinine (sCr), ureic-nitrogen (sBUN), albumin [sALB]) were assessed pre-, -post_0h_ and post_24h_. A principal component (PC) analysis was performed in each IMU spot to extract the main variables that could explain eWL variance. After extraction, PC factors were inputted in multiple regression analysis to explain biomarkers delta change percentage (Δ%). sCK, sCr, sBUN, sALB presented large differences (*p* < 0.05) between measurements (pre < post_24h_ < post_0h_). PC’s explained 77.5–86.5% of total eWL variance. sCK Δ% was predicted in 40 to 47% by L_1_–L_3_ and MP_left_; sCr Δ% in 27% to 45% by L_1_–L_3_ and MP_left_; and sBUN Δ% in 38%-40% by MP_right_ and MP_left_. These findings could lead to a better comprehension of how eWL (impacts, player load and approximated entropy) could predict acute kidney and muscle injury. These findings support the new hypothesis of mechanical kidney injury during trail running based on L_1_–L_3_ external workload data.

## 1. Introduction

Recently, inertial measurement units (IMU) composed by different microsensors (gyroscope, accelerometer and magnetometer) have been developed and used for the analysis of human movement [1]. In sports, these types of sensors have been used to quantify the external workload in different team and individual sports [2]. Although the external workload and gait biomechanics has traditionally analyzed under laboratory conditions using three-dimensional capture systems [3], this microsensors technology have been started to use for the analysis of external workload and biomechanical aspects for the improvement of optimal performance in laboratory conditions [4,5,6,7].

Nowadays, there has been a growing interest in quantifying workload in infield settings. In this sense, different researches have assessed by microtechnology the external workload in individual and team sports. The most common used variables were peak acceleration [8], impacts at different ranges [9], accumulated accelerometer load indexes [10,11,12,13] and specific events during competition and training sessions such as collisions, jumps or specific events among others [14,15,16,17]. In running sports, IMU sensors have been used for the analysis of velocity [18], stride length [18], vertical ground reaction forces [19], body segment kinematics [19], postural stability [4] multi-joints external workload [5,6] and peak accelerations [20], among others. Currently, the usefulness of this type of measurements of external workload has been questioned without having data on the impact they cause at the physiological level, which is why research has been carried out that analyzes both variables to have a clearer understanding of the physical demands and physiological together [21]. The understanding of both internal and external workload variables could better explain the mechanisms of physical damage.

Recently, studies have been carried out in individual and team sports on how this external workload affects muscle function and damage, related to impacts [22,23], jumps [24], changes of direction and speed [25,26]. This muscle damage has been quantified by biochemical (creatine kinase, lactate dehydrogenase, magnesium ant others) [22,27], functional [28] and perceptual methods [29]. On the other hand, the impact that the body receives on each action can also affect the renal function due to the constant mechanical trauma that the kidney can suffer during long-term and moderate to high intensity events, although there is insufficient evidence to relate the external workload and the possible mechanical trauma at the renal level [30]. The most common methods to quantify renal function are cystatin C (sCyst-C), serum creatinine (sCr), estimation of glomerular filtration rate (sGFR), creatinine clearance and blood ureic nitrogen (sBUN) [31,32,33]; in addition to other novel markers that could suggest subclinical injury as serum albumin (sALB), neutrophil gelatinase-associated lipocalin (NGAL) and kidney injury molecule 1 (KIM-1) [34].

Muscle and kidney injury in endurance athletes has been widely reported [35] and is a point of concern in the health of these types of athletes because the combination of factors such as workload, dehydration and heat strain can trigger acute kidney injury and cause future complications. It has been found that sports that cause a lot of eccentric actions, are carried out for prolonged hours and are exposed to adverse environmental conditions are those that are most likely to cause muscle and kidney injury, this is the case of trail running [33]. Due to the lack of information regarding the external workload indicators that could affect muscle and kidney injury, the purpose of the study was to explore which external workload factors have more influence on the responses of muscle and kidney injury biomarkers in experienced endurance trail runners.

## 2. Methods

### 2.1. Design

Participants were assessed -pre (serum test), during (physical external workload), -post_0h_ (serum test) and -post_24h_ (serum test) a trail running event. Participants were asked to run 3 × 11.76 km trail circuit (total distance: ~35.27 km, cumulative positive ascend: 1815 m [from 906 to 1178 m.a.s.l.]). The altimetry of the event and variables with its measurement time can be assessed at Figure 1. The thermal stress index (WetBulb-Globe Temperature [WBGT]) registered throughout the event was 23.2 ± 1.8 °C (temperature: 24.46 ± 2.42 °C and humidity 77.88 ± 10.91%) according to the WBGT (QuestTemp 36, 3M, MN, USA). Final running time was 290.3 ± 54.2 min.

### 2.2. Participants

A total of 20 male runners (age = 38.95 ± 9.99 years, weight = 71.94 ± 12.59 kg, height = 171.15 ± 9.52 cm) took part of the study. Participants were recruited among heat-acclimatized (life and train near event place), trained (running training = 533.1 ± 201.6 min/week) and experienced ultra-endurance runners (years of trail running experience = 6.3 ± 5.8 years). Participants who reported any muscular or metabolic diseases or recent (<6 months) physical injury of the lower limbs were excluded from the study.

Experimental protocol was approved by the Institutional Review Board (Reg. Code UNA-CECUNA-2019-P005). All the participants were informed of the details of the experiment procedures and the associated risks and discomforts. Each subject gave written informed consent, according to the criteria of the Declaration of Helsinki, regarding biomedical research involving human subjects (18th Medical Assembly, 1964, revised in 2013 in Fortaleza).

### 2.3. Material and Procedures

#### 2.3.1. Serum Markers

A 5 mL of blood was drawn from an antecubital vein directly into a blood collection sterile tube (BD Vacutainer^®^, New Jersey, NJ, USA) containing spray-coated silica particles activator and a gel polymer for serum separation. Samples were centrifuged at 2000 *g* relative centrifugal force (RCF) for 10 min using tube centrifuge (PLC-01, Gemmy Industrial Corp., Taipei, Taiwan). During sample collecting process, blood samples were stored on ice in a special cooler (45QW Elite, Pelican^TM^, California, CA, USA) until they were stored in a freezer (−20 °C) the same sample extraction day. Sample processing was performed a day after the event under controlled and isolated room using an automatic biochemical analyzer (BS-200E, Mindray, China) by photometry method.

The variables extracted from analysis were serum creatinine (sCr, mg/dL), serum creatine kinase (sCK, IU/L), serum ureic nitrogen (sBUN, mg/dL), serum albumin (sALB, IU/L). Delta percentage of change was calculated for each variable between pre- and -post_0h_ or -post_24h_. Kidney functional loss and Acute Kidney Injury (AKI) was considered and classified following Acute Kidney Injury Network (AKIN), the Risk, Injury, Failure, Loss of kidney Function, and End-stage kidney disease (RIFLE) [36] and the Kidney Disease Improving Global Outcomes (KDIGO) [37] criteria as follow: AKI_risk_ (sCr increase of 150% or acute increase or ≥0.3 mg/dL) and AKI_injury_ (sCr increase of 200%). Additionally, exertional rhabdomyolysis (ER) was considered if sCK level exceeded 1000 UI/L [38].

#### 2.3.2. Physical External Workload

To assess locomotion and kinematic variables, inertial measurement units (IMU) (WIMU PRO™, RealTrack Systems, Almería, Spain) were used in order to register the external workload during running. Six different IMU were attached at six different anatomical spots using a special spandex dark-suit (pat. pending) developed for the research. The suit was made with pockets for each IMU’s in six different spots (one IMU at T_2_–T_4_, one IMU at L_1_–L_3_; two IMU at right [VL_right_] and left [VL_left_] vastus lateralis muscle bellies and two IMU 3 cm cephalic to right [MP_right_] and left [MP_left_] malleolus peroneus) (see Figure 2a). Suit incorporated dark elastic straps were used to avoid vibrations or non-wanted movements of the devices during running (see Figure 2b).

The IMU’s were previously calibrated following protocols for this specific microsensor [39]. This IMU has been used for the assessment of neuromuscular running workload [6] and its reliability had been tested in it use, attached to different body places [39]. Total variables data extracted from IMU software were analyzed using a principal component analysis (PCA) in order to explain total variance [40].

Total variables analyzed were: Player Load per min (AU, PL/min), Player Load difference between segments (AU, PL_Dif_), approximated entropy (ApEn, AU), maximum acceleration (m/s^−1^, Acc_max_), total accelerations (Acc, n/min), total decelerations (Dec, n/min), average acceleration (Acc_avg_, m/s^2^), average deceleration (Dec_avg_, m/s^2^), maximum speed (Speed_max_, m/s), average speed (Speed_avg_, m/s), metabolic power (MP, W/kg), high metabolic load distance (HMLD, m/min), explosive distance (>16 km/h) (D_>16 km/h_, m/min), maximum heart rate (HR_max_, bpm), average heart rate (HR_avg_, bpm), total impacts (Impacts_total_, n/min), and total impacts at 1 g ranges from 0 to >30 g (Impacts_total_, n/min).

### 2.4. Statistical Analysis

All the variables were reported using the mean, standard deviation and lower and upper limits. Mean significant differences of serum tests variables were explored using a one-way analysis of variance and main differences between time measures were confirmed using Bonferroni method. The magnitude of the differences (effect size) was qualitatively interpreted using partial omega squared (ω_p_^2^) as follows: >0.01 small; >0.06 moderate and >0.14 large [41]. Change delta’s percentage (Δ*%*) was reported between time measurement in each variable as follow:$$\Delta \%=\frac{\mathrm{post}-\mathrm{pre}}{\mathrm{pre}}*100$$

From a total of 458 variables extracted from external workload assessment, only 169 max, average and relative variables were selected for correlation matrix exploration (31 for T_2_–T_4_, 20 for L_1_–L_3_, 24 for VL_right_, 24 for VL_left_, 35 for MP_right_ and 35 for MP_left_). A threshold of r < 0.7 was used as a criteria selection for extract non correlated variables for running each Principal Component Analysis (PCA). Selected variables were prior scaled and centered (*Z*-Score) and PCA’s were suitable considering Kaiser-Meyer-Olkin values (KMO = 0.61–0.635) and Barleth Sphericity test was significant (*p* < 0.01). After PCA, eingvalues greater than 1 were included for extraction in respective principal component (PC). An orthogonal rotation using VariMax method was used for identification of respective loadings in each PC, then only loadings greater than 0.6 were retained for interpretation and the highest loading was reported when a cross loading was identified between PC’s.

After PC’s were extracted, multiple lineal regressions (R^2^) were performed in order to analyze how the principal components found from each body segment explain both muscle and kidney injury markers change after the event. Alpha was prior set as *p* < 0.05. Data analysis was performed using the Statistical Package for the Social Sciences (SPSS, IBM, SPSS Statistics, v.22.0, Illinois, USA).

## 3. Results

### 3.1. Muscle and Kidney Injury Serum Markers

Table 1 shows the mean differences (lower and upper range) and changes of muscle and kidney injury serum markers by measure moment. Large effect size was found in all kidney (sCr, sBUN and sALB) and muscle (sCK) injury variables between pre- and -post_0h_ or -post_24h_. Regarding kidney injury, the greatest change was found between pre- vs. -post_0h_, while in muscle damage the highest change was shown between pre- vs. -post_24h_. A total of 4/20 (20%) cases met diagnosis criteria [38] for ER and 11/20 (55%) cases met diagnosis criteria [36,37] for AKI_risk_ and 3/20 (15%) AKI_injury_ based on sCr.

### 3.2. External Workload Variables Selected per Body Segment

The external workload variables outcome and principal components of each body segment spot is shown in Table 2. The body segments that explained the highest percentage of variance were MP of both legs and L_1_–L_3_, while the location that explained the lowest was VL_left_. Additionally, PL, PL_Dif_ between segments, ApEn, different levels of impacts and impacts_total_/min were the most common variables that explained total variance of the workload.

### 3.3. Prediction of Serum Change by External Workload Variables of Each Body Location

Finally, Table 3 shows the prediction of muscle and kidney injury serum variables change by the workload principal components of each body segment. At pre- vs. -post_0h_, the highest prediction values were found in: sCr by MP_left_ (45%) and L_1_–L_3_ (27%); sBUN by MP_right_ (40%) and MP_left_ (38%); and sCK by MP_left_ (47%) and L_1_–L_3_ (40%). At pre- vs. -post_24h_ the highest prediction values was found in: sCr by T_2_–T_4_ (74%); sBUN by MP_right_ (10%) and T_2_–T_4_ (10%); and sCK by L_1_–L_3_ (59%). sALB was not predicted by any of the workload variables.

## 4. Discussion

For our knowledge, this is the first research that explores and analyzes which external workload indicators influence the most on muscle and kidney injury biomarkers during endurance trail runners. It was found that muscle and kidney injury biomarkers presented large differences (ω_p_^2^: 0.17–0.53; *p* < 0.01) between measurements (pre < post_24h_ < post_0h_). Extracted PC’s explained 77.5 to 86.5% of total external workload variance. sCK Δ% was predicted in 40% and 47% by L_1_–L_3_ and MP_left_ PC’s respectively; sCr Δ% in a 27% and 45% by the L_1_–L_3_ and MP_left_ PC’s; and sBUN Δ% in 38% and 40% by MP_right_ and MP_left_ PC’s.

The sCK increased when pre vs. post_0h_ (Δ% = 322.56 ± 503; *p* < 0.01) and pre vs. post_24h_ (Δ% = 337.75 ± 303.25; *p* < 0.01) with post values between 691.05 and 680.87 UI/L after 35 km trail running with the presence some cases of exertional rhabdomyolysis (20% of total sample met diagnosis criteria). Same similar results that other studies when compared pre, post_0h_ and post_24h_ [42] is also reported in recent evidence after endurance running in 42.195 km marathon with 131,900 UI/L [43], 89.3 km with 5718-to-54,231 UI/L [44], 100 km ultramarathon with 200,000 IU/L [45], 161 km and 5500 m cumulative climbing with 38,218-to-95,940 UI/L [46] or 1550-to-264,300 UI/L [47].

When sCK rises above 1000 UI/L [38], this could lead to a condition known as exertional rhabdomyolysis, caused by the release of sarcoplasmic proteins into the bloodstream [48] due to the damage and disintegration of striated muscle during strenuous physical exertion [22,49,50]. In the present study 20% of participants presented exertional rhabdomyolysis. There could be observed as expected that when the distance of the event increases the sCK rises significantly, but trail running events tend to provoke greater sCK changes when compare to other road and flat events, due to the higher slope variations (uphill and downhill) [51]. This slope conditions could lead to significant higher impacts and metabolic workload than relative flat events as marathon [51], because the greater effort that should be made in order to absorb impact and constant slope changes that requires higher eccentric muscle contractions [52].

Not only muscle biomarkers have affected, kidney functional (sCr and sBUN) and subclinical (sALB) injury biomarkers also increased. sCr, sBUN and sALB obtaining the highest values -post_0h_. sCr and sALB not presented statistical differences between pre- and -post_24h_, but sBUN maintained equal values in post_0h_ and post_24h_ with large differences respect to pre-. This increase in serum kidney functional biomarkers as sCr have been previously reported in running endurance events during 42.2 marathon with 7.97 mg/dL [43], 89.3 km with 2.99–12.88 mg/dL [44], or 161 km with 5500 m cumulative climbing with 1.1–4.9 mg/dL [39] with clinical symptoms and hospitalization (acute renal dialysis and hyponatremia), or 100 km ultramarathon with 17.64 mg/dL [45] and 135 m ultramarathon with 1–1.34 mg/dL without any clinical symptoms [53].

The found acute rise in sCr (Δ% = 45.7, sCr difference ≥ 0.3 mg/dL) could be diagnosed as AKI_risk_ or AKI_injury_ following AKIN, RIFLE and KDIGO reference ranges (70% of total sample met criteria) [36,37,54]. Additionally, despite there is evidence that the rise in sCr or sBUN by itself could not be considered as AKI because there is no enough information of subclinical injury, the rise in sALB should suggest transitory functional loss due to tubular or glomerular damage [54]. Considering the above information and analyzing mean change of sBUN, sCr and sALB levels of the participants, this data suggest AKI presence after trail running due to functional, physiological and structural changes. The increase of this serum biomarkers levels could respond to mechanical muscle and kidney trauma, this last condition has been suggested in other contact sports [30] but not deeply explored in running sports until present analysis.

In order to explore the impact of the mechanical muscle and kidney trauma on serum biomarkers, a principal component analysis was performed for each body segment in order to select the main group of variables that could explain external workload variance. This statistical technique has been used in other sports as a data meaning method. PC’s extracted explained 77.5–86.5% of total external workload variance. In this study PL, total impacts, impacts at different ranges and entropy partly explained the total variance in all body segments. The difference in impacts range between segments is related to the ground-to-ground contact, finding the ranges of higher impacts in lower limb respect to lumbar region and back.

Real time monitoring using IMU devices have been used in order to explore fatigue, neuromuscular changes and physiological effects of running [22] but there is the first attempts to analyze multi segment external workload in trail running. Previous studies have analyzed multi-segments workload at laboratory conditions through player load and peak accelerations [8]. These studies found greater external workload in the nearest segment with the ground-to-ground contact (ankle) and at a faster speed. In trail running, only two previous researchers analyzed external workload during trail running at different points [51,55], but only one analyzed the workload dynamics during all the race in one body segment (T_2_–T_4_) through player load, metabolic power, entropy, speed, vertical stiffness and heart rate [51]. The highest values of external workload were found in downhill segments, while the highest metabolic response was found in uphill segments [56,57,58].

In this sense, if in downhill segments was found the greatest external workload, where eccentric muscle contractions are important for the technical abilities of the trail runner as changes of direction, accelerations and decelerations [51]; it is explainable that impacts, player load and entropy took part in all body segments PCs. The first two variables are calculated from the three axial accelerations of the human body [6,9] and entropy try to explain the regularity of signal dynamics [59]. In downhill segments, the gait biomechanics is more irregular, and the greatest fly time provokes that especially lower limb suffer greater impacts.

Despite the fact that there is enough evidence of the effect of endurance running and trail running on muscle damage [28,42], the rise in muscle damage serum concentration has not been associated with contextual factors as finish time, age, gender, delayed onset muscle soreness or running experience [47]. This could be due to the etiology of serum blood levels increasing is related to muscle damage during endurance running, compared to other activities [42,60]. This is why there is an increasing interest in exploration of which external workload variables could predict serum muscle and kidney changes. It was found that muscle and kidney mechanical trauma hypothesis theory [30] could be supported by the results of the present manuscript, where sCK Δ% was predicted in 40% and 47% by L_1_–L_3_ and MP_left_ PC’s respectively; sCr Δ% in a 27% and 45% by the L_1_–L_3_ and MP_left_ PC’s and sBUN Δ% in 38% and 40% by MP_right_ and MP_left_ PC’s. These results may suggest that core muscle resistance, optimal absorption forces, and efficient running economy could be protective factors to avoid greater muscle and kidney mechanical injury. Another consideration about the role of L_1_–L_3_ in impact absorption could develop mechanical kidney trauma due to kidney shaking and nephritis [30].

While the results of this study have provided valuable information about the influence of external workload variables and its potential prediction of muscle and kidney mechanical injury changes during endurance trail running, some limitations must be acknowledged. Due to the nature of trail running, it was difficulties in the assessment of some serum biomarkers in the middle of the race. Some serum biomarkers as myoglobin and Cyst-C as markers of kidney injury could be assessed in future studies as well as other novel subclinical AKI markers as NGAL and KIM-1 should be assessed in future studies in order to confirm kidney structural damage and early detect AKI [34,61]. Due to the organic exploration of the real setting conditions during and after running, some factors such food intake and liquid intake were no monitored and were led ad libitum. After running recovery strategies were no restricted but registered as internal control. As expected, these results must be addressed considering the specific anthropometric, experience and competitive level of analyzed sample, and should not be extrapolated to other populations that show different kinematical behavior because of their individual characteristics or competitive level.

Finally, future research could explore the impact of contextual factors as slope variations, age, finish time, carried weight during running, dehydration status and other contextual variables on muscle and kidney health during trail running. As well as more exploration of other conditions as medication, dehydration, heat strain and health status that could trigger a potential condition and future exertional rhabdomyolysis or AKI. There is a necessity of new evidence around the efficiency of recovery protocols to maintain or recover serum baseline levels.

## 5. Conclusions

Based on the previous results, it is confirmed that after an endurance trail running event, there are not only changes in muscle damage markers, but also produced changes in kidney injury biomarkers that could considered as a transitory loss of kidney function. These findings give new evidence that in order to a better understanding of muscle and kidney health, not only pre and post-race serum biomarkers have to be assessed, but other contextual factors such as locomotion variables, temperature, heat strain and dehydration should also be assessed in order to better understand the global muscle and kidney injury phenomenon during trail running.

Considering that this is the first study to address which variables explain the behavior of external workload total variance and it relation to changes in serum biomarkers of kidney and muscle injury; it is essential to indicate that in the case of endurance trail running, the external workload principal components predicts from 10% to 47% of the serum changes as sCK, sBUN and sCr values.

These findings could lead to a better understanding of how external workload could predict transitory acute kidney injury and exertional rhabdomyolysis in endurance sports. Variables as impacts, player load, approximated entropy and player load difference between segments should be assessed as external workload indicators of mechanical muscle and kidney injury. Additionally, this paper has contributed to a new hypothesis about muscle and kidney mechanical trauma in non-contact sports as trail running, due to high number and magnitude of ground reaction forces, change of direction, acceleration and deceleration involved during uphill and downhill running.

Considering there is enough evidence of the development of AKI during endurance sports and there is data in other populations of how cumulative AKI events could lead to future CKD, why should be different in athletes? Despite there is a lack of evidence around the long term effects of AKI in endurance runners and other kidney injury biomarkers should be assessed, practitioners should address this new information and take action around the optimal physical conditioning, hydration protocols, acclimatization and other considerations in order to reduce kidney injury risk during endurance events.

## Figures and Tables

**Figure 1 ijerph-16-03909-f001:**
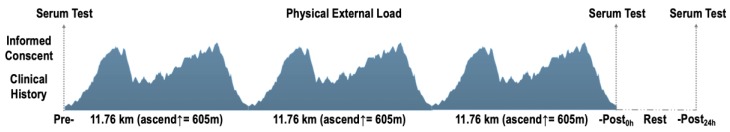
Schematic design of study variables with time measurement and trail altimetry.

**Figure 2 ijerph-16-03909-f002:**
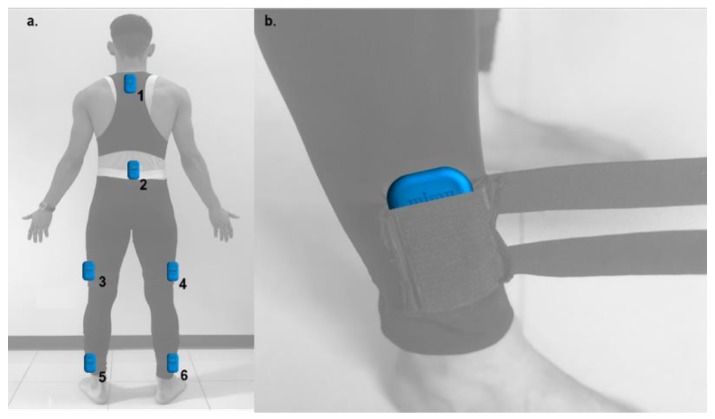
IMU device’s setting (**a**) 1: T_2_–T_4_, 2: L_1_–L_3_, 3: VL_left_ and 4: VL_right_ vastus lateralis, 5: MP_left_ and 6: MP_right_ malleolus peroneus; (**b**). body attachment anti-vibration straps system.

**Table 1 ijerph-16-03909-t001:** Mean differences (lower and upper limits) and change delta’s percentage in muscle and kidney injury serum makers by measure moment.

Category Variable	Pre-	-Post_0h_	-Post_24h_	F_(2.28)_ (*p*)	ω_p_^2^ Rating	Δ% Pre- vs. -Post_0h_	Δ% Pre- vs. -Post_24h_
Kidney Injury							
sCr (mg/dL)	1.22 ± 0.29 (0.66 to 1.7)	1.71 ± 0.4 (1.06 to 2.7) *	1.3 ± 0.29 (0.91 to 1.78) ^†^	19.05 (<0.01)	0.53 large	45.67 ± 42.26 (−1.49 to 171.21)	9.02 ± 12.74 (−14.93 to 31.58)
sBUN (mg/dL)	14.4 ± 4.42 (6 to 24)	19.92 ± 5.2 (8.7 to 29) *	18.88 ± 4.89 (13 to 27) *	14.004 (<0.01)	0.46 large	48.91 ± 68.05 (−15 to 323.1)	37.21 ± 37.41 (−35 to 116.67)
sALB (IU/L)	4.31 ± 1.22 (0.29 to 4.99)	5.01 ± 0.82 (1.71 to 5.84) *	4.67 ± 0.25 (4.16 to 5.06)	4.145 (0.027)	0.17 large	92.55 ± 362.99 (1.2 to 1634.48)	15.6 ± 59.74 (−10.1 to 230.71)
Muscle Damage							
sCK (IU/L)	274.5 ± 384.36 (45 to 1688)	691.05 ± 591.43 (229 to 2695) *	680.87 ± 552.07 (244 to 2400) *	11.021 (<0.01)	0.39 large	322.56 ± 503.01 (42.23 to 2371.1)	337.75 ± 303.25 (−4.56 to 976.23)

sCr: serum creatinine, sBUN: serum ureic blood nitrogen, sALB serum albumin and sCK: serum creatine kinase. * Significant differences with Pre- (*p* < 0.01); ^†^ Significant differences with -Post_0h_ (*p* < 0.01). Lower and upper limits were reported in brackets.

**Table 2 ijerph-16-03909-t002:** External workload variables outcome and extracted principal components of each body segment spot.

		Outcome, M ± SD (95%CI)	PC1	PC2	PC3	PC4
T_2_–T_4_	Eigenvalue		3.198	1.352	1.324	1.101
	% variance		35.53	15.02	14.71	12.24
	% cumulative variance		35.53	20.55	65.26	77.5
	PL_Dif_ T_2_–T_4_–L_1_–L_3_ (AU)	274.17 ± 251.37 (−306 to 654.67)	−0.818			
	ApEn (AU)	0.43 ± 0.1 (0.26 to 0.64)	0.81			
	Impacts_total_/min	314.77 ± 55.56 (201.29 to 417.54)		0.781		
	Acc_max_ (m/s^−1^)	4.41 ± 1.23 (3.19 to 7.21)		−0.624		
	PL/min (AU)	1.6 ± 0.57 (0.86 to 2.73)		0.766		
	Speed_max_ (m/s)	5.05 ± 0.85 (3.83 to 7.24)			0.938	
	Impacts_0-1 g_/min	108.06 ± 39.69 (23.3 to 171.77)				−0.892
L_1_–L_3_	Eigenvalue		3.186	2.481	1.458	1.234
	% variance		31.86	24.82	14.58	12.34
	% cumulative variance		31.86	56.68	71.26	83.60
	Impacts_total_/min	170.57 ± 36.64 (111.87 to 247.79)	−0.843			
	Impacts_0-1 g_/min	95.28 ± 38.02 (18.8 to 161.86)	−0.763			
	ApEn (AU)	0.51 ± 0.11 (0.24 to 0.76)	0.725			
	PL/min (AU)	2.78 ± 0.53 (2.16 to 3.88)	0.753			
	PL_Dif_ T_2_–T_4_–L_1_–L_3_ (AU)	274.17 ± 251.37 (−306 to 654.66)		0.847		
	Impacts_8-9 g_/min	2.16 ± 3.27 (0.32 to 14.68)		0.789		
	Impacts_6-7 g_/min	5.93 ± 2.13 (3.04 to 9.39)			−0.677	
	Impacts_1-2 g_/min	64.42 ± 13.25 (41.84 to 87.39)			0.781	
	Impacts_5-6 g_/min	10.18 ± 3.39 (5.46 to 16.54)				0.933
VL_right_	Eigenvalue		1.606	1.24	1.081	-
	% variance		32.12	24.81	21.61	-
	% cumulative variance		32.12	56.93	78.55	-
	PL/min (AU)	3.96 ± 0.93 (2.47 to 5.62)	0.823			-
	Impacts_7-8 g_/min	3.22 ± 1.1 (1.68 to 5.45)	0.892			-
	Impacts_3-4 g_/min	10.74 ± 4.2 (4.72 to 21.75)		0.931		-
	PL_Dif_ VLright–MPright (AU)	52.3 ± 435.04 (−1045.51 to 1002.44)			−0.642	-
	ApEn (AU)	0.46 ± 0.11 (0.32 to 0.67)			0.838	-
VL_left_	Eigenvalue		1.951	-	-	-
	% variance		65.05	-	-	-
	% cumulative variance		65.05	-	-	-
	PL/min (AU)	3.88 ± 0.88 (2.7 to 5.74)	−0.696	-	-	-
	Impacts_5-6 g_/min	5.57 ± 2.04 (2.49 to 10.06)	0.796	-	-	-
	Impacts_3-4 g_/min	11.43 ± 4.98 (3.72 to 21.48)	0.913	-	-	-
MP_right_	Eigenvalue		2.614	1.978	1.238	1.092
	% variance		32.67	24.72	15.48	13.64
	% cumulative variance		32.67	57.4	72.87	86.52
	Impacts_8-9 g_/min	5.77 ± 2.04 (3.11 to 12.12)	0.859			
	PL/min (AU)	4.52 ± 1.03 (3.16 to 6.56)	0.73			
	ApEn (AU)	0.36 ± 0.18 (0.04 to 0.81)	−0.862			
	Impacts_total_/min	115.84 ± 27.91 (74.29 to 163.89)		0.867		
	PL_Dif_ VL_right_–MP_right_ (AU)	52.3 ± 435.04 (−1045.51 to 1002.44)		0.94		
	Impacts_1-2 g_/min	28.83 ± 10.77 (10.43 to 49.95)		0.779		
	Impacts_6-7 g_/min	6.95 ± 2.35 (3.32 to 12.12)			−0.845	
	Impacts_3-4 g_/min	9.28 ± 4.25 (4 to 16.34)				0.95
MP_left_	Eigenvalue		2.538	2.206	1.58	1.175
	% variance		28.2	24.51	17.55	13.06
	% cumulative variance		28.2	52.71	70.26	83.32
	PL_Dif_ VL_left_–MP_left_ (AU)	193.75 ± 0.88 (−1228.3 to 935.59)	−0.766			
	Impacts_6-7 g_/min	7.12 ± 2.74 (1.94 to 11.42)	0.754			
	Impacts_8-9 g_/min	5.47 ± 1.61 (2.7 to 8.5)	0.888			
	PL/min (AU)	4.53 ± 1.07 (2.95.1 to 7.18)		−0.903		
	Impacts_4-5 g_/min	6.9 ± 2.63 (1 to 11.46)		0.887		
	Impacts_1-2 g_/min	26.58 ± 8.12 (13.11 to 45.84)			0.88	
	Impacts_total_/min	155.46 ± 20.87 (98.33 to 184.1)			0.842	
	Impacts_3-4 g_/min	10.46 ± 4.01 (2.38 to 18.28)				0.92

Note. M: mean; SD: standard deviation; CI: confidence interval; AU: arbitrary units; PC: principal component.

**Table 3 ijerph-16-03909-t003:** Body segments external workload indicators (principal components) that predicted muscle and kidney injury serum changes.

Δ% Pre- vs. -Post_0h_				
Body Segment	sCr	sBUN	sALB	sCK
T_2_–T_4_	R^2^ = 0.23, β = 44.03	R^2^ = 0.22, β = 51.91	R^2^ = 0.18, β = 100.55	R^2^ = 0.14, β = 333.97
	*p* < 0.01 **	*p* < 0.01 **	*p* = 0.3	*p* = 0.025 *
L_1_–L_3_	R^2^ = 0. 27, β = 45.36	R^2^ = 0.2, β = 55.99	R^2^ = 0.29, β = 112.16	R^2^ = 0.4, β = 350.02
	*p* < 0.01 **	*p* = 0.014 *	*p* = 0.286	*p* = 0.019 *
VL_right_	R^2^ = 0.11, β = 42.69	R^2^ = 0.33, β = 47.97	R^2^ = 0.36, β = 101.28	R^2^ = 0.33, β = 336.79
	*p* < 0.01 **	*p* < 0.01 **	*p* = 0.223	*p* = 0.01 **
VL_left_	R^2^ = 0.07, β = 41.63	R^2^ = 0.10, β = 48.39	R^2^ = 0.16, β = 96.09	R^2^ = 0.2, β = 324.57
	*p* < 0.01 **	*p* < 0.01 **	*p* = 0.25	*p* < 0.01 **
MP_right_	R^2^ = 0.2, β = 45.22	R^2^ = 0.4, β = 51.51	R^2^ = 0.44, β = 119.34	R^2^ = 0.36, β = 373.01
	*p* < 0.01 **	*p* = 0.019 *	*p* = 0.243	*p* = 0.024 *
MP_left_	R^2^ = 0.45, β = 47.33	R^2^ = 0.38, β = 50.39	R^2^ = 0.45, β = 96.35	R^2^ = 0.47, β = 335.28
	*p* < 0.01 **	*p* < 0.01 **	*p* = 0.202	*p* < 0.01 **
Δ% Pre- vs. -Post_24h_				
Body Segment	sCr	sBUN	sALB	sCK
T_2_–T_4_	R^2^ = 0.74, β = 877.57	R^2^ = 0.1, β = 39.17	R^2^ = 0.29, β = 22.41	R^2^ = 0.3, β = 363.58
	*p* = 0.02 *	*p* < 0.01 **	*p* = 0.265	*p* < 0.01 **
L_1_–L_3_	R^2^ = 0.45, β = 5.07	R^2^ = 0.19, β = 37.14	R^2^ = 0.32, β = 19.62	R^2^ = 0.59, β = 493.04
	*p* = 0.229	*p* = 0.057	*p* = 0.529	*p* < 0.01 **
VL_right_	R^2^ = 0.19, β = 6.95	R^2^ = 0.22, β = 36.82	R^2^ = 0.18, β = 19.53	R^2^ = 0.22, β = 324.08
	*p* = 0.077	*p* < 0.01 **	*p* = 0.325	*p* < 0.01 **
VL_left_	R^2^ = 0.002, β = 7.46	R^2^ = 0.08, β = 34.9	R^2^ = 0.13, β = 13.87	R^2^ = 0, β = 189.63
	*p* = 0.039 *	*p* < 0.01 **	*p* = 0.41	*p* = 0.967
MP_right_	R^2^ = 0.56, β = 4.53	R^2^ = 0.23, β = 31.26	R^2^ = 0.5, β = 9.65	R^2^ = 0.27, β = 473.47
	*p* = 0.207	*p* = 0.126	*p* = 0.69	*p* = 0.015 *
MP_left_	R^2^ = 0.12, β = 10.13	R^2^ = 0.1, β = 39.93	R^2^ = 0.08, β = 13.14	R^2^ = 0.39, β = 413.12
	*p* = 0.025 *	*p* < 0.01 **	*p* = 0.524	*p* < 0.01 **

* *p* < 0.05, ** *p* < 0.01.

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
