# Peer review of "External Workload Indicators of Muscle and Kidney Mechanical Injury in Endurance Trail Running"

_ijerph, 2019, doi:10.3390/ijerph16203909_

Round 1

Reviewer 1 Report

This paper presents endurance trail running indicators to muscle and kidney mechanical injuries. According to this research, it would be the possibility to reduce acute kidney injury risk during endurance events. Manuscript contributed to could lead understanding how influences, player load and could predict transitory kidney injury and rhabdomyolysis. The work raises questions for future research to optimize physical condition, correct hydration protocols. Moreover, it put more attention to acclimatization and for considerations to reduce acute kidney injury risk during endurance events.

The manuscript can be published.

Author Response

Dear reviewer:

We have carefully considered all your comments in order to improve the paper (ijerph-618885). Please find enclosed our detailed answers to your queries. Please find some changes to the initial manuscript following both reviewers comments. 

Reviewer 2 Report

In this manuscript, authors demonstrated that physical external workload which was measured by inertial measurement unit was associated with increased levels of serum creatinine (Cr) and creatine kinase (CK) in well-trained trail runners. This study seems to be well-designed and the subject is interesting. However, there are some concerns in this study. The reviewer’s comments are described as follows.

1. Increased Cr levels after intensive exercise are generally considered not pathological process but physiological response. Definition of acute kidney injury (AKI) is based on the fact that elevation of serum creatinine 0.3 mg/dl was related to increased inhospital mortality. Transient increased creatinine in athletes is never associated with mortality as well as transition to chronic kidney disease. Therefore, the term AKI seems to be inappropriate. Acute Dialysis Quality Initiative suggested to distinguish “functional” with “subclinical” AKI. If authors would like to suggest clinically significant AKI in this study, they should present data of serum markers of renal tubular injury, including NGAL and KIM-1 along with creatinine and blood urea nitrogen. Or, authors should replace AKI with simply “increased Cr”.

2. As with AKI, the term rhabdomyolysis also seems to be inappropriate. This medical term means symptomatic process and is associated with organ damage. The reviewer disagrees that exercise-related transient elevation of serum CK was called as rhabdomyolysis in this study. As authors know, elevation of CK is substantially different between individuals. If authors would like to suggest the clinically significant rhabdomyolysis, they should evaluate muscle injury by clinical findings (grading) along with serum CK levels. Or, authors should replace the term rhabdomyolysis with “increased CK”.

3. In Table 3, is R2 = 11.5 in the relation between MPleft and serum Cr correct? The value seems to be too much.

Author Response

Dear reviewer:

We have carefully considered all your comments in order to improve the paper (ijerph-618885). Please find enclosed our detailed answers to your queries. Please find some changes to the initial manuscript following both reviewers comments. 

1-2. Thanks for your positive feedback. Based on your review, we have made some changes that really helped to improve the quality of the manuscript.
You can see in text that we have made changes regarding the use of the term of rhabdomyolysis and we have carefully use the term AKI when necessary considering both functional and subclinical AKI.

In this line, the term rhabdomyolysis was change for exertional rhabdomyolysis considering that in this case the cause of these conditions was understood as muscle damage and no other clinical conditions as described previously:

Clarkson, P. M. (2007). Exertional Rhabdomyolysis and Acute Renal Failure in Marathon Runners: Sports Medicine 37, 361–363. doi:10.2165/00007256-200737040-00022.

DeFilippis, E. M., Kleiman, D. A., Derman, P. B., DiFelice, G. S., and Eachempati, S. R. Spinning-induced Rhabdomyolysis and the Risk of Compartment Syndrome and Acute Kidney Injury: Two Cases and a Review of the Literature. SPORTS HEALTH, 3.

Dekeyser, B., Schwagten, V., and Beaucourt, L. (2009). Severe rhabdomyolysis after recreational training. Emergency Medicine Journal 26, 382–383. doi:10.1136/emj.2008.065771.

Patel, D. R., Gyamfi, R., and Torres, A. (2009). Exertional Rhabdomyolysis and Acute Kidney Injury. The Physician and Sportsmedicine 37, 71–79. doi:10.3810/PSM.2009.04.1685.

We added information regarding the conclusions based on functional and sub-clinical kidney injury considering the rise in sALB as a biomarker that suggest tubular damage. This represented  an opportunity to expand the explanation around the rise in sALB as a indicator of tubular and glomerular damage following recommended references: McCullough, P. A., Shaw, A. D., Haase, M., Bouchard, J., Waikar, S. S., Siew, E. D., … Ronco, C. (2013). Diagnosis of Acute Kidney Injury Using Functional and Injury Biomarkers: Workgroup Statements from the Tenth Acute Dialysis Quality Initiative Consensus Conference. Contributions to Nephrology, 13–29. doi:10.1159/000349963.

This supported the idea that subclinical AKI should be considered as AKI despite there is no kidney dysfunction, this condition was assessed by sALB in this manuscript. Other functional damage was assessed through the rise in sCr and sBUN. Considering this information, we have presented evidence of both functional and structural damage, understood as subclinical and physiological damage, both considered as AKI.

Ronco, C.; Kellum, J.A.; Haase, M. Subclinical AKI is still AKI. Crit. Care 2012, 16, 313.

Limitation was expanded due to the lack of markers as neutrophil gelatinase-associated lipocalin and kidney injury molecule 1 as subclinical as early AKI indicators.  

Additionally, we understand that the diagnostic criteria and specific considerations about AKI have raised doubts. As you mention there are efforts to qualify these black spots in the understanding of this phenomenon. These concerns are further exacerbated in athletic populations, who have protective factors at a physiological level greater than other populations at possible risk (e.g. previous pathologies). In fact, recent published studies that classified functional and clinical AKI recognized that there is a need of new evidence in order to select the better diagnosis criteria.

Despite AKI clinical diagnosis criteria is not specific for athletes, there is recent evidence that have suggested AKI in endurance runners due to the rise in sBUN, sALB and sCr:

Boulter, J., Noakes, T., and Hew-Butler, T. (2011). Acute renal failure in four Comrades Marathon runners ingesting the same electrolyte supplement: coincidence or causation? S Afr Med J 101, 876–8.

Belli, T., Macedo, D. V., de Araújo, G. G., dos Reis, I. G. M., Scariot, P. P. M., Lazarim, F. L., et al. (2018). Mountain Ultramarathon Induces Early Increases of Muscle Damage, Inflammation, and Risk for Acute Renal Injury. Front. Physiol. 9. doi:10.3389/fphys.2018.01368.

Clarkson, P. M. (2007). Exertional Rhabdomyolysis and Acute Renal Failure in Marathon Runners: Sports Medicine 37, 361–363. doi:10.2165/00007256-200737040-00022.

DeFilippis, E. M., Kleiman, D. A., Derman, P. B., DiFelice, G. S., and Eachempati, S. R. Spinning-induced Rhabdomyolysis and the Risk of Compartment Syndrome and Acute Kidney Injury: Two Cases and a Review of the Literature. SPORTS HEALTH, 3.

Dekeyser, B., Schwagten, V., and Beaucourt, L. (2009). Severe rhabdomyolysis after recreational training. Emergency Medicine Journal 26, 382–383. doi:10.1136/emj.2008.065771.

Hoffman, M. D., and Weiss, R. H. (2016). Does acute kidney injury from an ultramarathon increase the risk for greater subsequent injury? Clin J Sport Med 26, 417–422. doi:10.1097/JSM.0000000000000277.

Hou, S.-K., Chiu, Y.-H., Tsai, Y.-F., Tai, L.-C., Hou, P. C., How, C.-K., et al. (2015). Clinical Impact of Speed Variability to Identify Ultramarathon Runners at Risk for Acute Kidney Injury. PLOS ONE 10, e0133146. doi:10.1371/journal.pone.0133146.

Junglee, N. A., Di Felice, U., Dolci, A., Fortes, M. B., Jibani, M. M., Lemmey, A. B., et al. (2013). Exercising in a hot environment with muscle damage: effects on acute kidney injury biomarkers and kidney function. Am. J. Physiol. Renal Physiol. 305, F813-820. doi:10.1152/ajprenal.00091.2013.

Kao, W.-F., Hou, S.-K., Chiu, Y.-H., Chou, S.-L., Kuo, F.-C., Wang, S.-H., et al. (2015). Effects of 100-km Ultramarathon on Acute Kidney Injury. Clinical Journal of Sport Medicine 25, 49. doi:10.1097/JSM.0000000000000116.

Mccullough, P. A., Chinnaiyan, K. M., Gallagher, M. J., Colar, J. M., Geddes, T., Gold, J. M., et al. (2011). Changes in renal markers and acute kidney injury after marathon running: Kidney Injury after Marathoning. Nephrology 16, 194–199. doi:10.1111/j.1440-1797.2010.01354.x.

Patel, D. R., Gyamfi, R., and Torres, A. (2009). Exertional Rhabdomyolysis and Acute Kidney Injury. The Physician and Sportsmedicine 37, 71–79. doi:10.3810/PSM.2009.04.1685.

Shimizu, Y., Takaori, K., and Maeda, S. (2017). Exercise-induced acute renal failure in a trainee cyclist without hypouricemia: Successful athletic career post-treatment. J Gen Fam Med 18, 432–435. doi:10.1002/jgf2.108.

This evidence have suggested that despite AKI in athletes is a transitory condition in response to muscle and mechanical damage, AKI should be considered because it presents the regular clinical criteria for classifications such as AKIN, RIFLE and KDIGO and there are some functional changes that should be addressed and it is sufficient to trigger attention protocols. sALB was a factor previously considered as a structural (glomerular and tubular) damage.

We really think that the analysis of sCr, sBUN in addition with sALB could give a better perspective of both functional and structural acute kidney damage.

Also, we were careful in the suggestion that AKI could develop future CKD because the lack of evidence in this affirmation.

Please see the changes made inside the manuscript.

3. Thank you for your comment. We correct the mistake 11.5 to 0.12. Please see correction in text.

We really appreciate your comments. We want to thank you for all your specific comments that have improved the quality of the manuscript and to allow us to better explain both structural and functional kidney damage using sALB, sCr and sBUN global analysis.

Round 2

Reviewer 2 Report

Authors have appropriately addressed the reviewer's concerns. The paper can be accepted.